# Sustainable Fish Feeds with Insects and Probiotics Positively Affect Freshwater and Marine Fish Gut Microbiota

**DOI:** 10.3390/ani13101633

**Published:** 2023-05-14

**Authors:** Imam Hasan, Simona Rimoldi, Giulio Saroglia, Genciana Terova

**Affiliations:** 1Department of Biotechnology and Life Sciences, University of Insubria, Via Dunant, 3-21100 Varese, Italy; ihasan@uninsubria.it (I.H.); genciana.terova@uninsubria.it (G.T.); 2Medical Devices Area, Institute of Digital Technologies for Personalized Healthcare-MeDiTech, Scuola Universitaria Professionale della Svizzera Italiana, Via La Santa 1, CH-6962 Lugano, Switzerland; giulio.saroglia@supsi.ch

**Keywords:** metagenomics, DNA barcoding, rainbow trout, European sea bass, firmicutes, Actinobacteria, Proteobacteria, *Lactobacillus*, *Bacillus*, *Aeromonas*

## Abstract

**Simple Summary:**

One of the greatest challenges to achieving a sustainable aquaculture is finding alternatives to fishmeal as a primary protein source in aquafeeds. Insects represent one of the most promising alternatives being explored and produced as replacements for this ingredient. This review addresses the use of two insect species (black soldier fly, *Hermetia illucens*, and yellow mealworm, *Tenebrio molitor*) in freshwater and marine fish diet formulations and the effect of insect meal on fish gut microbiota. Furthermore, the effects of a probiotic, namely, *Lactococcus lactis* subsp. *lactis*, are considered. The study of fish gut microbiota is very important for aquaculture practice as gut microbiota plays a significant role in nutrition metabolism, also affecting a number of other physiological functions, including fish growth and development, immune response, and pathogen resistance. Along with recent and promising results in this field, new insights and future directions on fish gut microbiota research are highlighted.

**Abstract:**

Aquaculture is the fastest-growing agricultural industry in the world. Fishmeal is an essential component of commercial fish diets, but its long-term sustainability is a concern. Therefore, it is important to find alternatives to fishmeal that have a similar nutritional value and, at the same time, are affordable and readily available. The search for high-quality alternatives to fishmeal and fish oil has interested researchers worldwide. Over the past 20 years, different insect meals have been studied as a potential alternate source of fishmeal in aquafeeds. On the other hand, probiotics—live microbial strains—are being used as dietary supplements and showing beneficial effects on fish growth and health status. Fish gut microbiota plays a significant role in nutrition metabolism, which affects a number of other physiological functions, including fish growth and development, immune regulation, and pathogen resistance. One of the key reasons for studying fish gut microbiota is the possibility to modify microbial communities that inhabit the intestine to benefit host growth and health. The development of DNA sequencing technologies and advanced bioinformatics tools has made metagenomic analysis a feasible method for researching gut microbes. In this review, we analyze and summarize the current knowledge provided by studies of our research group on using insect meal and probiotic supplements in aquafeed formulations and their effects on different fish gut microbiota. We also highlight future research directions to make insect meals a key source of proteins for sustainable aquaculture and explore the challenges associated with the use of probiotics. Insect meals and probiotics will undoubtedly have a positive effect on the long-term sustainability and profitability of aquaculture.

## 1. Introduction

Aquaculture is one of the fastest-growing food production sectors in the world, supplying more than half of the global fish supply. By 2050, it is expected that global aquaculture consumption will double. To guarantee long-term food security, efficient and sustainable animal production methods are urgently required. Aquafeeds are mainly based on fishmeal (FM) and fish oil (FO), the most abundant dietary protein source. However, the global increase in aquaculture production has required alternative feedstuff, which often has a detrimental effect on the growth, intestinal health, and immune response of farmed marine fish [1,2,3].

One problem the aquaculture feed industry needs to solve is that of replacing FM with other protein sources. If this cannot be done, serious concerns exist about the industry’s capacity to remain economically and environmentally stable. Sardine, anchovy, herring, capelin, mackerel, and other forage and small pelagic marine fish species are sources of FM and FO. However, owing to the gradual decline in wild marine fish stocks [4,5,6,7,8,9,10,11,12,13,14], it will soon no longer be viable to use these aquafeed raw materials. Plant-based proteins and oils comprise the primary substitutes for FM and FO due to their greater availability and lower cost [4]. Indeed, soybean and other protein- and lipid-rich plants have replaced FM and FO in farmed fish diets [5,6,7,8,9,10]. Soybean meal is a top-rated source of protein in plant-based diets. However, plant-based diets may decrease fish growth and disease resistance due to anti-nutritional substances in plant meals that affect fish feed intake, digestion, and nutrient utilization, causing inflammation in many fish species’ intestines [4,5,6,7,8,9,10,11,12,13,14,15,16,17,18]. Furthermore, plant-based feed ingredients are deficient in protein, lack a balanced amino acid profile, are unpalatable, and compete with other food industrial sectors [16,19,20].

As a result of all of these factors, the need to find better ways of getting protein from other valuable alternatives to FM, for example, animal feed ingredients such as by-products from slaughterhouses or insect meals (IM), has increased [18,21,22]. From this perspective, insects have the potential to open a new world of sustainable, protein-rich ingredients for aquafeeds. Furthermore, single-cell proteins (SCP) deriving from microalgae, bacteria, and yeast are also being used as fish feed ingredients. Insect, micro- and macroalgae, and microbial meals are becoming more popular as aquafeed components [23,24,25]. Among the scientific publications currently available in indexed databases, just 19% were focused on IM, whereas 16% discussed the use of microalgae: IM and microalgae are both components that offer much promise in the aquafeed sector. Animal health and metabolism are influenced by a complex relationship between the host, the gut microbiota, and their feed. A balanced microbiota is important for the overall health and well-being of the host. The fish gut microbiota significantly affects fish health and physiology [8]. It helps develop the immune system and promotes nutrient utilization [26,27]. The structure of the gut microbial communities, including microbial diversity, is highly influenced by the ingredients of the diet because the microbiota reacts quickly to dietary changes [28]. It is well established that replacing FM with plants, yeast, IM, or animal by-products influences the biodiversity and number of gut bacteria [22,29,30,31,32,33,34]. Furthermore, the bioactive compounds present in insects can alter the complex communities of intestinal microbiota. Consequently, the variety and richness of fish gut bacteria have changed as a result of replacing FM in their diets with IM, either from *Hermetia illucens* (HI) or *Tenebrio molitor* (TM) [35,36,37].

The gut microbiota is usually called an “extra organ” because of its significant role in many physiological processes of the host, including digestion, metabolism, reproduction, development, and immunological response. In recent years, new alternative components have been investigated and used in aquafeeds. Since gut microbes are important for digestion and health, a number of studies have been carried out to determine how diet affects the gut microbiota of aquatic species. Therefore, this review highlights current developments and future perspectives of alternatives to conventional protein sources, that is, how IMs used as aquafeed ingredients affect the gut microbiota of marine fish. Furthermore, we review the effects of probiotics on fish intestinal microbiota.

## 2. Insects for Sustainable Aquaculture

Insects are an environmentally sustainable protein-rich feed ingredient for farmed fish. IM is being considered a potential alternative to FM as a protein source in aquaculture feeds. Interest in using IM as an alternative to FM has increased since the European Union (EU) authorized the use of IM from seven distinct insect species in aquaculture feeds [38,39]. According to the circular economy concept, insects are worthy candidates for aquafeed ingredients. Many country’s aquaculture industries increasingly depend on IM instead of FM. Insects are nutritionally valuable due to their high protein (60–80%), fat (31–43%), essential amino acids, and mineral and vitamin content [24]. Due to their high protein content and balanced amino acid profile, IM has emerged as a popular alternative to FM and a new source of protein in terrestrial and aquatic animal diets [40,41]. Therefore, insects constitute an excellent alternative to conventionally produced animal-based protein sources for use as feed [42,43]. Many studies have investigated the effects of FM/IM substitution in various fish species diets. The European Commission has withdrawn the ban on using processed animal proteins generated from insects in aquafeed for farm fish under regulation EU-2017/893. As a result, IM can now be used in aquafeeds. The regulation lists the seven types of insects that are allowed: black soldier fly, *Hermetia illucens*; common housefly, *Musca domestica*; yellow mealworm *Tenebrio molitor*; lesser mealworm, *Alphitobius diaperinus*; house cricket, *Acheta domesticus*; banded cricket, *Gryllodes sigillatus*; and field cricket, *Gryllus assimilis*. Of these, flies in particular have been the focus of aquafeed industry research in recent years owing to their many advantages over other animal protein sources [44]. HI and TM are the main species presently receiving considerable attention for aquaculture feed formulations [45]. Most studies have shown that replacing FM with IM is a good approach to increase aquaculture sustainability; however, the results vary based on the fish and insect species used. We recently obtained promising findings in marine and freshwater carnivorous fish species with the dietary use of different inclusion rates of black soldier fly and yellow mealworm meals [36,37,46,47,48,49]. Table 1 represents the research that our group has done on the effects of IM on fish gut microbiota.

### 2.1. Black Soldier Fly (Hermetia illucens, HI)

When producing IM, the black soldier fly HI is an excellent potential species because its amino acid profile is similar to that of FM, making it a suitable alternative protein source [24]. HI is the most widely studied and used insect species, representing our research group’s primary alternative to FM raw material. Indeed, HI can be raised quickly, have a high fertility rate, and turn waste into high-quality protein [50]. An increasing number of feeding trials have been conducted, demonstrating that HI meals can be a suitable FM replacement in aquaculture diets [41,44,51]. During the last few decades, approximately 130 research publications with the terms “Black soldier fly,” “Larvae meal,” and “Aquaculture” have been indexed in PubMed, Scopus, Web of Science, and other databases. Prepupae of HI comprise an intriguing choice for producing IM since mass-rearing procedures for high-quality output currently exist [24]. Using HI in fish feed provides a way to solve problems in the aquaculture industry related to managing a sustainable aquatic environment. According to several studies, HI can replace conventional FM and totally replace SBM in aquaculture feeds without negatively influencing fish growth, feed efficiency, digestion, or fillet quality [37,52,53,54]. Our experiments have shown that rainbow trout (*Oncorhynchus mykiss*) can tolerate up to 50% HI meal in their diet with no negative effects on fish growth and survival [36,37,46,48,55] and with positive effects on the gut microbiota of fish.

#### Effects of FM/HI Meal Replacement on Fish Gut Microbiota

HI meals are becoming more popular in aquaculture feeds, but ideal inclusion levels still must be determined to ensure fish growth and health. An increasing number of studies have examined the effects of substituting HI meals for FM in the diets of different species of fish. Most research recommended partial replacements of FM with HI meals. However, some recent studies revealed 100% replacement without affecting fish growth, especially for carnivorous fish [52].

Regarding fish growth, health, and gut microbiome, our group’s work has shown that partial or up to 50% inclusion of HI meal in the diet is well tolerated and has no negative effects on fish growth or survival. Diet has a significant role in shaping the gut microbiota, but the surrounding environment and environmental factors can also significantly impact microbiota composition. Our research group previously evaluated the effects of different HI inclusion levels in high-FM diets on fish gut microbiota using high-throughput sequencing technologies [36,37,46]. In all the experiments, we applied high-throughput sequencing of the 16S rRNA gene to assess the dynamics of major gut bacterial taxa in response to diet. PICRUSt1 bioinformatics software was used to determine gut microorganisms’ key active biological pathways. We reported that the partial substitution of dietary FM with 10%, 20%, or 30% of a defatted HI meal had an important effect in modulating the intestinal transient (allochthonous) and resident (autochthonous) bacterial communities in trout [36,37,46].

HI diet increased butyrate-producing bacteria in the fish gut [36,37,49] and led to diversification and other alterations in the intestinal bacterial makeup of rainbow trout [37,53,56]. In addition, dietary IM increased the colonization of beneficial bacteria, such as lactic acid bacteria (LAB), which are often used as probiotics in animal nutrition [36,37]. This was a good result as it is known that beneficial bacteria species compete with gut detrimental bacteria for niche space and produce and secrete antimicrobial peptides, thus protecting the host from colonization and proliferation of environmental pathogens [57].

Based on the metabarcoding results, three phyla, Firmicutes, Proteobacteria, and Tenericutes, were found to be the most abundant in the digestive tract of rainbow trout [37], and in fish fed with 10–30% HI meal; diversity was higher in allochthonous, but not in autochthonous gut microbiota [36,37]. Instead, another study [53] observed that trout fed a diet containing 20% HI meal had a higher species richness in their gut microbiota. Furthermore, the autochthonous bacterial community significantly influenced host metabolism and health status more than the allochthonous intestinal bacteria.

Fish gut microbiota studies vary in many ways, including the techniques used to analyze the microbiome. The dietary HI meal’s effects on autochthonous microbiota of trout were first explored using the gradient gel electrophoresis (DGGE) method [53], which identified a lower number of bacterial species than the Illumina MiSeq method, which we used in all our studies [36]. We analyzed the inclusion of 10%, 20%, and 30% HI meals on the autochthonous intestinal microbiota of rainbow trout (*O. mykiss*) and found a reduced abundance of Proteobacteria and an increased abundance of *Mycoplasma*, which produce lactic and acetic acid as final products of its fermentation [36,37]. These differences in the composition of the autochthonous intestinal microbiota are due to the prebiotic characteristics of fermentable chitin. In one of our previous studies [37], Proteobacteria, Firmicutes, and Actinobacteria dominated trout’s allochthonous gut microbial community. Interestingly, also other studies reported that LAB (Firmicutes phylum) were only found in large numbers in the gut contents of trout that had been fed IM but they were absent in gut mucosa [53]. In contrast, trout intestinal mucosa in our study contained many Proteobacteria (Gammaproteobacteria) bacteria, which was in line with previous work on rainbow trout [53,58]. The most common phyla are not the only ones for which differences between these findings and our previously published data were observed. Fish mucosa samples contained considerably fewer operational taxonomic units (OTUs) (74 vs. 450, respectively) than fish gut digesta samples [37]. These results agree with another study [59], which found that microbial diversity was lower in the gut mucosa than in the luminal part. This indicates that certain species of bacteria colonize the gut mucosal layer poorly and that the number of bacteria and the diversification of the autochthonous bacterial community may be different from the allochthonous microbiota [60]. In our studies, 20% IM increased biodiversity (Shannon and Simpson evenness indices) but not bacterial richness [36,37]. In line with previous research, we found that HI meal inclusion in the trout diet had positive effects on gut bacterial biodiversity [37,53,56]. Furthermore, since dietary effects may in part be biased by taxa from the feed microbiome [61], we included the feed as control and did not use digesta as a proxy for the intestinal microbiome [36,37]. Indeed, to fully unveil the response of gut microbiota to dietary changes, we performed concurrent profiling of feed microbiota, and digesta- and mucosa-associated gut microbiota.

In addition to their protein and fat content, insects contain a large amount of chitin, which is the building material that gives strength to the exoskeletons of insects. Studies have shown that the gut microbiota of fish may be altered by chitin [62]. In Atlantic salmon, a chitin-rich diet altered gut microbiota, revealing over 100 autochthonous bacterial species [63]. Dietary chitin or chitosan modulates fish gut microbiotas due to its prebiotic, antibacterial, and immunomodulatory properties [37,46,49,64,65]. Many fish cannot digest chitin, so it is possible to consider it as an insoluble fiber with possible prebiotic qualities. These properties may help maintain a well-balanced and healthy gut microbiota. The gut microbiota helps digest otherwise indigestible feed ingredients, generating short-chain fatty acids (SCFAs), which are the main energy source for intestinal epithelial cells [66]. Furthermore, our latest research [47] on the effects of chitin-rich shrimp head meal (SHM) and HI pupal exuviae on the gut microbiota of rainbow trout demonstrated that HI exuviae exert a modulatory influence on the fish gut microbiota by increasing the number of Firmicutes and Actinobacteria. Pupal exuviae thus represent a promising prebiotic for fish gut microbiota, increasing gut bacterial richness and the amount of beneficial chitin-degrading bacteria, such as *Bacillus* species, which promotes SCFA synthesis, especially butyrate. Similarly, adding krill or chitin into salmonid diets increased bacterial alpha diversity [62]. Therefore, our findings should not be unexpected when considering the chitin level of the IM.

Chitin is a prebiotic that increases the diversity of the bacteria in the gut. A healthy gut is typically characterized by a diverse bacterial population. In contrast, decreased diversity is typically associated with dysbiosis and illness risk, due to low bacterial competition for space and resources and enteric pathogen colonization [67,68]. The addition of HI meal to the trout diet significantly decreased indigenous Proteobacteria in the intestinal digesta [36,37]. The same finding was obtained in a study on the digesta and mucosa-associated trout microbiota [56]. Chitin, an insoluble fiber, may reduce Proteobacteria in IM-fed groups. According to several investigations, chitin and deacetylated chitin derivatives are antibacterial and bacteriostatic against Gram-negative pathogens [47,69]. We reported that trout-fed HI meal showed decreased Gammaproteobacteria, including the genera *Shewanella*, *Aeromonas*, *Citrobacter*, and *Kluyera*, which are considered responsible for some diseases in fish [36]. Therefore, including IM meal in trout diets has a positive effect that inhibits potential pathogen growth. Fish fed 20% and 30% HI diets had more *Mycoplasma*-genus bacteria in their intestines, and these may be beneficial [36,37,47]. Many studies have identified *Mycoplasma* as the predominant genus in the distal intestines of rainbow trout and other farmed salmonids [33,70,71]. Bacilli and Clostridium, which are also included in the phylum Firmicutes, are closely related to *Mycoplasma*. They are generally obligate symbiotic microbes of the gastrointestinal ecosystem because their small genome size makes it unlikely that complex metabolic functions take place in the fish gut [36]. Lactic and acetic acids are the main metabolites of *Mycoplasma* bacteria [71,72]. *Mycoplasma* maintains intestinal homeostasis in trout by using fermentable substrates and releasing end products from bacterial fermentations [73]. Recent research on trout revealed that a lower level of *Mycoplasma* in the gastrointestinal tract makes the fish more susceptible to disease [74]. These findings suggest that *Mycoplasma* produces antimicrobial chemicals, such as lactic and acetic acids, which are the main metabolites that benefit host health.

### 2.2. Yellow Mealworm (Tenebrio molitor, TM)

Yellow mealworms are becoming more popular as an alternative source of protein in aquaculture diets due to their high efficiency in converting organic waste, being considered an ideal circular economy insect. Defatted TM provides up to 63.84% crude protein and an amino acid composition similar to that of FM [75]. Furthermore, TM contain anti-tumoral, antibacterial, antioxidant, and immunomodulatory, physiologically active compounds [76,77]. TM has been evaluated as a potential alternative to FM as a protein source in the diets of various fish species. The nutritional value of TM varies with its substrate composition and rearing settings. Although most studies have indicated that 25% to 30% of TM be included in the diet [78], rainbow trout fed different FM/TM meal replacement levels showed better performance [49]. Significant growth improvement was seen in red seabream (*Pagrus major*) fed diets containing 65% defatted TM larval meal, completely replacing FM [79]. TM showed the highest apparent digestibility coefficient of the four IMs tested in Nile tilapia [80]. This proves that TM larvae can replace FM as a protein source in fish diets. One of our studies examined the impact of replacing FM with TM meal in rainbow trout diets on fish weight gain and gut and skin microbiome [49]. Dietary FM substitution with TM has been explored extensively on fish development performance but less on host symbiotic microbial population [49]. Like HI, TM contains bioactive chemicals that are abundant in chitin and lauric acid and affect the gut microbiota [35,36,37]. Most current research on fish microbiota has focused on the bacterial diversity that may be discovered in the fish’s gut; however, fish also have distinct microbial diversity in other important body sites. Particularly, the skin microbiota of fish and most farm animals has not been thoroughly studied but would require careful consideration. Fish skin constitutes one of their vital mucosal barriers to the outer world. Thus, skin microbiota plays a very important role in preventing fish diseases. In one of our studies, therefore, we investigated how the gut and skin microbiota of trout changed when FM was replaced with TM larvae meal [49].

#### Effects of FM/TM Meal Replacement on Fish Gut Microbiota

A considerable amount of research has been conducted on mealworm meals in aquafeeds. TM is an excellent alternative to FM, positively influencing fish growth rates and gut microbiota. The appropriate TM meal inclusion rates in feeds for different fish species depend on the nutritional requirements of a given fish species and the nutritional quality of the TM, which in turn depends on the diet and culture conditions of the larvae. Insect meal manufacturers have increased defatted insect meal production in recent years. Defatting insect meal increases crude protein and degradation resistance [24]. In rainbow trout diets, 25% or 50% TM meal did not affect fish weight but significantly improved feed conversion and the protein efficiency ratio [81,82]. The amount of protein, amino acids, micronutrients, lipids, and fatty acids in TM meal makes it a suitable replacement for FM in aquafeeds based on its effects on fish growth performance. In contrast, 50% of full-fat TM diets in European seabass reduced fish growth compared to FM diets [83]. In marine carnivorous fish species, high TM levels in aquafeed have led to reduced growth [84]. Our study found no statistically significant differences in growth performance features in rainbow trout after 90 days of feeding with either a 100% substitution of FM with a partially defatted TM diet or a diet without TM [49].

Many studies have focused on the impact of substituting TM meal for FM on growth and development without attempting to understand the processes that underlie these effects. It is important to use molecular genetics and genome sequencing to determine how TM meal works, how it is metabolized, and how it is absorbed by the digestive systems of different cultured fish species [49]. The gut microbiota plays an important role in enhancing feed digestion, which benefits the general health of fish [85]. As TM is being used in fish diets as a raw material, it is important to understand how gut microbes respond to adding TM to the diet. Several fish species, including rainbow trout, have been investigated to determine how dietary TM affects the composition and diversity of gut microbiota [35,49,86]. Our research group demonstrated how 100% of TM influences rainbow trout gut microbial populations [49]. Substituting FM with TM meal did not influence the species richness and variety of gut mucosal bacteria [50], a finding similar to that obtained from our previous studies [36,37]. Consistent with our findings, feeding rainbow trout (*O. mykiss*) or sea trout (*Salmo trutta m. trutta*) a hydrolyzed TM meal diet did not affect digesta-associated bacteria [86,87]. According to the results of our metagenomic analysis, the phylum Tenericutes was most represented in trout intestine irrespective of diet, followed by Proteobacteria and Firmicutes in descending order [49]; all these bacteria taxa play a key role in the host’s nutrition and metabolism. Furthermore, the abundance of *Lactobacillus* and *Enterococcus* bacteria increased in the intestines of juvenile rainbow trout fed a diet containing with 20% TM meal [86]. The prebiotic characteristics of chitin in dietary IM may be responsible for the increase in lactic acid bacteria. However, in our study on 100% FM substitution with a partially defatted TM diet, intestinal LAB did not increase [49]. This was a surprising result, especially compared to what we had seen in the intestines of trout-fed diets with HI meal [36,37]. Indeed, substituting FM with IM from HI larvae positively modulated rainbow trout gut microbiota by raising the levels of LAB, which are helpful bacteria commonly used as probiotics in the diet of fish and other vertebrates [36,37,53]. There is no doubt that LAB is crucial for degrading dietary fiber. In addition, they actively participate in host defense against pathogenic organisms by generating bactericidal chemicals, such as lactic acid, hydrogen peroxide, bacteriocins, and biosurfactants, which inhibit pathogen colonization of the intestinal epithelium [88,89]. The relative abundance of Actinobacteria increased in the digestive tracts of trout when TM larvae meal was added to their diet, but this effect was not evident in European sea bass or gilthead sea bream [35]. Indeed, the gut microbiota is usually changed towards Firmicutes and/or Actinobacteria when dietary fiber such as chitin is included [46,49,53,90]. Taken together, our data revealed that there were no negative effects on rainbow trout intestinal microbiota populations when FM was completely replaced with TM. No noticeable dysbiosis symptoms were found, but only slight microbial changes were seen [49]. The research revealed that TM larvae meal is a valid substitute for FM as an animal protein in aquafeeds.

## 3. Probiotics for Sustainable Aquaculture

Probiotics are living microorganisms that, when administered correctly, positively regulate an organism’s health [91]. They are regarded as important modulators of many biological processes such as digestion, immunological activation, restoring microbial balance, and modulating the microbiota composition and have potent antioxidant qualities due to their effects on the gut microbiota [92]. Probiotics can inhibit pathogens in various ways, including by directly competing for nutrients and cell attachment space and generating inhibitory molecules, such as lactoferrin, lysozyme, bacteriocins, siderophores, and enzymes. Probiotics secrete proteases, amylases, and lipases that degrade those feed ingredients the fish gut cannot digest, leading to enhanced growth and nutrient conversion efficiency [93,94,95]. In aquaculture, many probiotic microbial strains are now used [96]. LAB, such as *Lactobacillus* sp., *Bacillus* sp., *Enterococcus* sp., and yeast, *Saccharomyces cerevisiae*, are the most common probiotics used in aquaculture [97,98]. These microorganisms are widely distributed in nature in the digestive tracts of farmed fish and regulate the fish microbiota as permanent or transitory inhabitants [99]. Probiotics in aquaculture are generally also used to reduce antibiotic use and promote aquaculture industry sustainability. The misuse of antibiotics has a negative effect on the aquatic environment, particularly in aquatic ecosystems where antimicrobials can persist for a long time and help bacteria become resistant to multiple antibiotics [100]. Antibiotics also help fish grow but their use as growth promoters has reduced the variety and abundance of indigenous gut microbiota, negatively affecting fish immune systems [101]. For these reasons, antibiotics have been restricted in farmed animals in the EU since 2006 [102,103]; therefore, several research projects have attempted to substitute antibiotics with probiotics to help the growth and development of farmed animals [104].

Probiotics boost feed digestibility and nutrient absorption in cultured fish, leading to better fish growth and conversion rates [105]. They also maintain gut microbiota balance, especially at larval stages, when vaccination is challenging [101]. Probiotics are also being used more in aquaculture, and studies have confirmed the advantages for commercially important farmed fish [106,107,108]. In our recent work, gilthead sea bream fed low and high dosages of probiotic *Lc. lactis* subspecies *lactis* showed higher weight gain than control fish fed a diet without probiotics [109]. High-throughput sequencing was used in our study to analyze the alterations in sea bream gut microbial populations after *Lc. lactis* subsp. lactis feeding. The findings here indicate that digestion and nutrient utilization had improved in gilthead sea bream fed probiotics. The same results were seen when *Lactobacillus* spp. and *Shewanella putrefaciens* Pdp11 were administered to gilthead sea bream [110]. Many other farmed fish species showed improved growth performance when *L. lactis* was used as a probiotic [111,112,113]. Concerning the microbiota analysis, we also analyzed the microbiota populations associated with feeds at the end of our feeding trial to determine how stable the probiotics were in the fish diets. Firmicutes and Proteobacteria represented the most numerous bacterial phyla, followed by the Bacteriodetes and Fusobacteria in descending order. Compared to the most representative genera of Firmicutes phylum, the relative abundance of the probiotic *L. lactis* was higher.

The proportion of *L. lactis* included in the control diet was 0%, while it was 64% and 71% in the treatment diets, respectively, which corresponds with administering a low and high dose of probiotics to fish [109]. Gilthead sea bream fed high dosages of *Lc. lactis* had an increase in Spirochete bacteria phylum in their gut, which were almost absent in fish fed low doses *of Lc. lactis* and in the control fish. Around 200 different genera in the Firmicutes phylum, including *Lactobacillus*, help maintain fish intestine health [37,109,114]. Commensal Firmicutes and Bacteroidetes produce butyrate, acetate, and propionate SCFAs by dietary fiber fermentation. The gut microbiota of sea bream on high-probiotic diets had a Proteobacteria/Firmicutes ratio five times higher than in the other groups. This result is not surprising since *Lc. lactis* subsp. *lactis* produces the antibiotic nisin, displaying strong activity against Gram-positive bacteria, and a vast majority of Firmicutes are Gram-positive [109,115].

In our study, the analysis of gut-adherent (autochthonous) microbiota showed a lack of colonization of the probiotic *Lc. lactis* in the host’s intestinal mucosa [109]. This result was expected because it is well-recognized that the underlying mechanisms of establishing probiotics in the host intestinal mucosa are challenging and influenced by complex molecular interactions. Our probiotic modified the fish gut microbiota without colonizing the host’s intestinal mucosa, proving that colonization is not always required to trigger host modification [109]. In terms of diversity indices, the analyses of intestinal microbiota also found significant and controversial differences between groups of fish. There was a significant difference in the variety and diversification characterized by alpha diversity parameters in fish fed a low-probiotic diet compared to the control or high-probiotic diet fish [109].

Consistent with our findings, the bacterial diversity in the intestinal mucosa of Atlantic salmon supplemented with LAB was higher [116]. On the other hand, in the probiotic-rich diet group, gut bacteria diversity was lowest despite reaching the highest growth rates. A functionally unbalanced ecosystem may reduce competition for opportunistic or invading bacteria if bacterial diversity decreases and is generally regarded as a negative outcome [110,117,118]. While it has been documented that administering prebiotics (specialized plant fibers that stimulate the growth of healthy bacteria) increases the microbial richness of the gut, evidence of the benefits of probiotics on fish remains less clear. According to findings in the literature, the dietary probiotic *Bacillus subtilis*, alone or in combination with prebiotics or microalgae, decreased gilthead sea bream species richness and diversity indexes [110,119]. Moreover, probiotics such as LAB produce antimicrobial substances that limit the growth of other microbes, which can change the gut microbiota’s composition and biodiversity [120].

The correlations found in the aforementioned studies between diet and fish gut microbiota suggest that well-designed probiotics could provide a potential way to improve fish growth performance and digestive ability. However, traditional probiotics, such as lactic acid bacteria and yeasts are not the dominant indigenous microbes in the digestive tract of fish, and their use in fish may risk causing microbial dysbiosis in some cases [121]. Therefore, identifying commensal beneficial bacteria in fish is of great value for the development of novel probiotics for aquaculture [121].

## 4. Metagenomic Analysis for the Identification of Gut Microbiota

Different culture-dependent methods followed by identification based on biochemical and phenotypic characteristics of bacteria were used to identify and characterize fish microbiota in previous times. Unfortunately, culture-dependent techniques give a limited picture of intestinal microbiota because only a low fraction, down to about 1% of the bacteria from fish intestine, can be cultivated [22]. Therefore, culture-independent molecular technologies, such as next-generation sequencing (NGS) technologies, targeted amplicon sequencing of the 16S rRNA gene [122,123,124], polymerase chain reaction-denaturing gradient gel electrophoresis (PCR-DGGE) [125,126,127], or 16S rDNA PCR-DGGE and RNA polymerase β-subunit gene quantitative PCR [128], have been used more recently to evaluate the intestinal microbiota. New research approaches have boosted our understanding of the interplay between microbes and their hosts. In particular, NGS enabled the identification and quantification of fish gut bacteria at unprecedented resolution, providing novel insights into the role of the microbiota in fish growth and health [66,129].

Metagenomics has thoroughly changed the study of the fish gut microbiota. With these methods, it is possible to directly look at the genome of microorganisms from samples taken from the environment [130,131]. It can provide a deeper understanding of the information that the retrieved DNA reveals about the host or environment-specific host species and help researchers understand microbial diversity in aquaculture. The use of 16S rRNA sequencing as the gold standard for identifying variability of the 16S rRNA to assess the composition of whole bacterial communities through culture-independent methods is being used by many researchers [59,132,133]. Additionally, metagenomics methods have been successfully used to find novel genes and microbial pathways as well as to discover functional dysbiosis [134].

To investigate the effect of aquafeed ingredients on gut microbiota composition, we used the Illumina MiSeq platform (Illumina, Italy) for high-throughput sequencing of the 16S rRNA gene (Figure 1) to analyze and characterize the complete gut microbiome of different fish species [22,36,37,46,47,48,49]. Using the bioinformatics application PICRUSt (Phylogenetic Investigation of Communities by Reconstruction of Unobserved States), the active biological pathways of gut bacteria could be identified. This method is being used more frequently in fish studies. It can quickly and cost-effectively capture detailed sequencing data that provides additional information on even minute amounts of bacteria [66]. The Illumina, Roche 454, and Ion Torrent PGM (Personal Genome Machine) are the three leading platforms used to study fish gut microbiota [135]. Many studies have been performed using these platforms on the microbiota that live in different parts of a fish’s body, such as the gastrointestinal tract, gills, and skin that evolve to permit colonization on the mucosal surfaces by complex commensal microorganisms [136]. The skin, gills, and digestive tract of fish are the main entry routes for pathogens [136]. Thus, most research on microbial communities has focused on these body regions. Owing to the development and continuous growth of aquaculture, microbiota study has recently increased. More focus is now being directed to the microorganisms that inhabit the gastrointestinal systems of different finfish and crustaceans, such as prawns and crabs [136]. Furthermore, there is an increasing interest in the impact of aquafeed on the fish gut microbiome, as some recent findings [61] are consistent with a model wherein gut microbial profiles are to a different degree influenced by bacterial DNA present in the feed itself through a “feed microbiome” carry-over effect.

## 5. Conclusion and Future Prospects for Applied Research

This review summarizes the results of studies from our research group on the effects of diets formulated to contain IM from either *T. molitor* or *H. illucens* to replace dietary FM on freshwater fish gut microbiota. It also emphasizes the connection between diet and fish gut microbiota, suggesting that “tuning” the microbiota composition through the use of new raw materials could offer a promising strategy towards a sustainable aquaculture. The effects of a probiotic (*Lactococcus lactis* subs *lactis*) used as a feed supplement on marine fish gut microbiota are also reviewed. According to metagenomic data, IMs from *H. illucens*, or *T. molitor* constitute valid alternative protein sources that can affect gut microbiota composition and overall fish health. With regard to *Lc lactis*, although this probiotic did not colonize the host’s intestinal mucosa, it positively influenced the fish gut microbiota by changing the abundance of different beneficial bacterial taxa and by impacting many metabolic pathways associated with protein absorption and digestion. Therefore, insects and probiotics used in fish diets have a positive effect on the composition of gut microbiota and on fish nutritional physiology.

We believe this information will be helpful to researchers and aquaculture experts in fish nutrition, particularly when developing novel feed formulations or experimenting with various feed components and additives.

## Figures and Tables

**Figure 1 animals-13-01633-f001:**
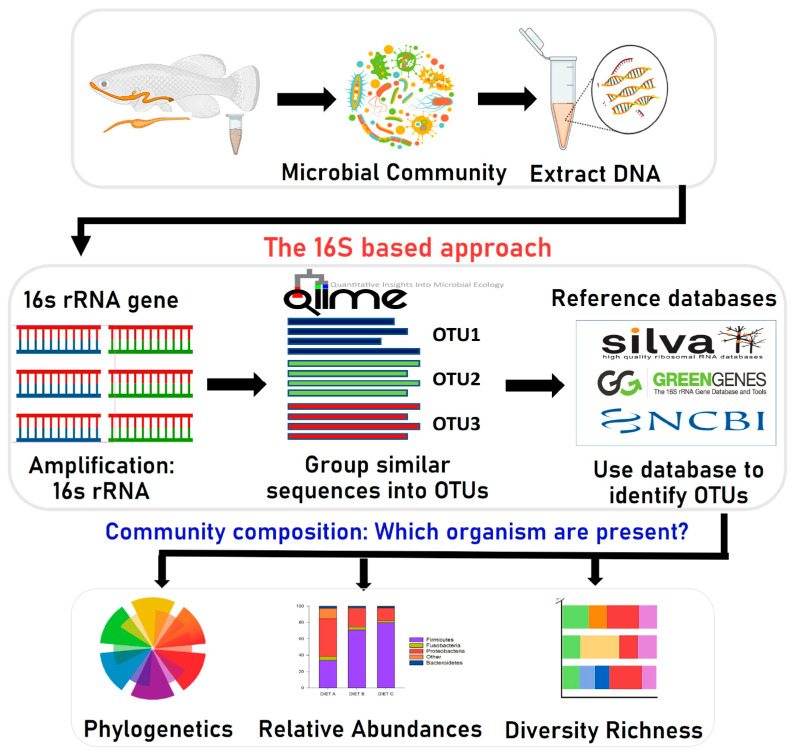
Overview of bioinformatics methods for metagenomic analysis of gut microbiota based on the 16S rRNA gene sequencing. OTU; Operational Taxonomic Unit.

**Table 1 animals-13-01633-t001:** Studies evaluating the effects of insect meal on fish gut microbiota using Next Generation Sequencing platforms.

Fish Species	FM Replacer in the Feed	Type of Sample	Sequencing Platform/16S rRNA Gene Region	Major Fnding(s)	Reference
Rainbow trout	Pupal exuviae meal *(Hermetia illucens)*	Intestinal mucosa and fecal matter	Illumina MiSeq: V4	*Hermetia illucens*-derived exuviae improved bacterial species richness in fish gut microbiota by increasing bacteria belonging to the Firmicutes and Actinobacteria phyla. Exuviae meal increased number of beneficial chitin-degrading bacteria, such as *Bacillus* genera, thus promoting the microbial synthesis of short chain fatty acids, primarily butyrate.	[47]
Rainbow trout	IM *(Tenebrio molitor)*	Intestinal mucosa	Illumina MiSeq: V3–V4	Rainbow trout gut and skin microbiota changed following FM/IM substitution. FM substitution with IM did not have negative effects on rainbow trout gut and skin microbial populations.	[49]
Rainbow trout	IM *(H. illucens)*	Intestinal mucosa and fecal matter	Illumina MiSeq: V4	Firmicutes, especially Bacilli, increased, whereas Proteobacteria, mainly *Pseudomonas*, decreased. *Lactobacillus* and *Bacillus* genus bacteria increased in the gut of fish fed with the IM based diet, but *Aeromonas* genus bacteria decreased dramatically in the same fish group.	[46]
Rainbow trout	IM *(H. illucens)*	Intestinal mucosa and fecal matter	Illumina MiSeq: V3–V4	The intestinal bacterial community of trout was influenced by IM, which improved fish gut health. Fish fed with insect-based diet showed higher bacterial diversity and less Proteobacteria than fish fed on an FM-based diet.	[36]
Rainbow trout	IM *(H. illucens)*	Digesta	Illumina MiSeq: V3–V4	IM increased fish gut microbiota biodiversity and richness. Insect-based feed boosted lactic acid and butyrate-producing bacteria, improving fish gut health.	[37]

## Data Availability

The authors declare that no new data were created.

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
