# Peer review of "Sustainable Fish Feeds with Insects and Probiotics Positively Affect Freshwater and Marine Fish Gut Microbiota"

_animals, 2023, doi:10.3390/ani13101633_

Round 1

Reviewer 1 Report

The paper is well written however it comes across as primarily a review and comparison of one lab groups research papers and how their results compare to the rest of published literature available at the time. There is very little discussion of results in general for use of insect meal and/or probiotics and overall trends, everything is posited as a comparison to the author’s research group and their results. As is this seems like a limited review and comparison of one group’s extensive research and a rehash of their already published results. While ample references are utilized, I would suggest expanding the discussion to be more inclusive of all results, even if at a general level, as opposed to the limited comparison of this groups results compared to a handful of other individual studies. There is also a high level of repetition of basic facts such as details about IM being suitable for FM replacement and why, as well as the general effects of probiotics on gut microbiomes. These sections could be greatly shortened to make room for some more generalized review discussion of overall trends observed across species and life stages with IM and probiotic inclusions.

Suggest to add “insects” and “probiotics” to the title somehow to help focus the title on the subject. As is, the title makes it sound like a potentially very broad review, but it is nicely focused on the inclusion and effects of insect meals.

Paragraphs starting on lines 53 and 61 are somewhat repetitive, suggest combining them and reducing repetition.

Paragraph starting on line 101 is extremely repetitive, even citing the EU authorization change twice, please reduce the similar statements throughout the paragraph.

English is overall relatively correct, only suggestion is to review tense of verbs, phrasing, and unneeded words in sentences throughout. The concepts and objectives of most sentences are obvious, but there are just some minor errors that would improve quality if corrected.

Reviewer 2 Report

The title of the manuscript "Sustainable fish feeds: a mini review focusing on the effects on gut microbiota of freshwater and marine fish" implies a broader approach than what was actually delivered by the authors. While the use of insect meal in aquaculture is promising, there is a dearth of studies that specifically address the effects on fish gut microbiota when insect meal is the sole ingredient targeted for review. Additionally, the majority of the review focused on insect meal does not address the effects on gut microbiota, but rather on performance parameters and substitution levels compared to traditional protein sources.

The authors were ambitious in the range of topics addressed, but limited in the scope of the topics included in this "mini-review." It is inadequate to assume that the effects of "Sustainable fish feeds" on gut microbiota can be limited solely to the use of insect meal and probiotics. There are various other types of additives, such as carbohydrates, fibers, and enzymes, as well as animal and plant-based ingredients, that can significantly impact fish gut microbiota.

It is suggested that the authors redefine the scope of the review and make explicit the central question they seek to answer. A review should address a specific question and not simply present a selection of articles related to the topic. Moreover, it is evident that the authors predominantly presented articles with positive outcomes of using insect meal and probiotics. This may indicate publication bias and potential skewness in the selected works.

Reviewer 3 Report

its not acceptable in present form as still needs alot of information to be included and refinement and arrangement of the text for better readability.. and review report is attached for the needful.

it needs a through revision for gramatical, phrases and effective language because it is an review article.

Round 2

Reviewer 2 Report

While I appreciate the changes made to the manuscript, I do not find the work to be sufficiently novel or original for publication. Moreover, a mini-review based on a comparison of previous studies from the authors' research group with the current literature is not a strong reason to publish a review. In fact, this comparison was likely made when the original studies were published. Therefore, I maintain my opinion to reject the study for publication.

Author Response

Authors: We thank the referee for his/her comments. Please consider that the MS comes across as primarily a mini-review and comparison of our lab groups research papers and how our results compare to the rest of published literature currently available. In the revised version, we have made a lot of changes including the title to help focus it on the subject. As it was, the title made it sound like a potentially very broad review, but it is nicely focused on the inclusion and effects of insect meals and probiotics. Furthermore, we have made substantial changes to the MS and included a Table and a Figure to better expand the discussion, as requested by the referee. For instance, in the new Table, the major findings of our research on gut microbiota have been listed. Please see all the changes in the file “Marked changes to MS”.